# Effect of Mg Content on Microstructure and Properties of Al–Mg Alloy Produced by the Wire Arc Additive Manufacturing Method

**DOI:** 10.3390/ma12244160

**Published:** 2019-12-11

**Authors:** Lingling Ren, Huimin Gu, Wei Wang, Shuai Wang, Chengde Li, Zhenbiao Wang, Yuchun Zhai, Peihua Ma

**Affiliations:** 1College of Metallurgy, Northeastern University, Shenyang 110000, China; renlingling27@126.com (L.R.); wangshuai106123@126.com (S.W.); lichengde20031698@126.com (C.L.); zhaiyc@smm.neu.edu.cn (Y.Z.); mapeihuadbdx@126.com (P.M.); 2Inner Mongolia Metal Research Institute, Baotou 014000, China; wwneu@hotmail.com; 3North East Industrial Materials & Metallurgy Co., Ltd, Fushun 113000, China; zhenbiaowang@hotmail.com

**Keywords:** arc additive manufacture, Al–Mg alloy, Mg content, microstructure, mechanical properties

## Abstract

In this study, an Al–Mg alloy was fabricated by wire arc additive manufacture (WAAM), and the effect of Mg content on the microstructure and properties of Al–Mg alloy deposits was investigated. The effects on the deposition surface oxidation, geometry, burn out rate of Mg, pores, microstructure, mechanical properties and fracture mechanisms were investigated. The results show that, when the Mg content increased, the surface oxidation degree increased; a “wave”-shaped deposition layer occurred when the Mg content reached 8%. When the Mg content was more than 6%, the burning loss rate of the Mg element increased significantly. With the increase of Mg content, the number of pores first decreased and then increased, and the size first decreased and then increased. When the Mg content reached 7% or above, obvious crystallization hot cracks appeared in the deposit bodies. When the Mg content increased, the precipitated phase (FeMn)Al_6_ and β(Mg_2_Al_3_) increased, and the grain size increased. When the Mg content was 6%, the comprehensive mechanical properties were best. The horizontal tensile strength, yield strength and elongation were 310 MPa, 225 MPa and 17%, respectively. The vertical tensile strength, yield strength and elongation were 300 MPa, 215 MPa and 15%, respectively. The fracture morphology was a ductile fracture.

## 1. Introduction

Wire arc additive manufacture (WAAM) is an additive manufacturing technology which is based on the discrete additive forming principle to form 3D physical parts suitable for the rapid manufacturing of medium and large-scale parts with medium complexity [1,2,3]. The WAAM method cannot realize net forming at present, which requires subsequent machining. At present, most WAAM aluminum alloys are Al–Cu alloys and Al–Si–Mg alloy, which need a solid solution and aging heat treatment to be strengthened [4,5,6]. In actual production, parts—especially large parts—will undergo severe deformation after quenching treatment, which makes the product accuracy difficult to control and subsequent machining difficult. Therefore, the pursuit of an Al–Mg alloy with excellent mechanical properties without heat treatment and strengthening has attracted the attention of WAAM manufacturing technology researchers [7,8].

At present, the research into WAAM Al–Mg alloys is not in-depth and is limited to traditional brands. Jiang [9] studied the rapid forming process of 5356 aluminum alloy based on CMT (Cold Metal Transfer) and proposed the anisotropy of mechanical properties, but did not explain the reason behind this. Horgar et al. [10] prepared AA5183 aluminum alloy by using the short pulse arc additive manufacturing method, but the mechanical properties of the deposits were not high, and the tensile strength and yield strength were 293 MPa and 145 MPa, respectively. Geng et al. [11] fabricated 5A06 aluminum alloy with a GTAW (gas tungsten arc welding) additive and obtained deposits with poor properties. The tensile strength, yield strength and elongation were 273 MPa, 124 MPa and 34%, respectively. As Mg is the main strengthening element of the Al–Mg alloy, it has high activity and is easy to burn and oxidize; thus, Mg content exerts an important influence on the performance of WAAM Al–Mg alloy deposits.

In this paper, Al–Mg alloy deposits with different Mg contents were prepared by the WAAM method, and the surface oxidation degree, geometric morphology, burning loss of Mg elements, pores, microstructure, mechanical properties and tensile fracture mechanism of Al–Mg alloy deposits were studied, laying a foundation for the further study of WAAM Al–Mg alloys.

## 2. Experimental Method

The Al–Mg alloy welding wire used in this paper was provided by North East Industrial Materials & Metallurgy Co., Ltd. (Fushun, China) and had a diameter of 1.2 mm. In this experiment, four kinds of welding wires with different Mg contents were prepared, and the target mass percentages of Mg content were 5%, 6%, 7% and 8%, with corresponding numbers of 1#, 2#, 3# and 4#, respectively. The Mg content mentioned in this paper was the target mass percentage of wire, and the measured chemical compositions of wires are shown in Table 1. The chemical composition of the deposit was measured from the upper, middle and lower points of the deposit and is shown in Table 2. The CMT + Advance forming process in Fronius Advanced CMT [12] power supply is adopted, and the equipment is shown in Figure 1. Because the Mg element is active, the Al–Mg alloy is greatly affected by process parameters (such as interpass temperature, etc.), and the experimental results are obtained under specific process parameters. The deposition process parameters are shown in Table 3, and the size of the deposition body is 200 mm × 150 mm.

The sampling location of the deposition body and the specification of tensile samples are shown in Figure 2. Two tensile test samples perpendicular to the deposition direction (horizontal samples) and two tensile test samples parallel to the deposition direction (vertical sample) were extracted from each deposit body. Tensile samples were processed by using a milling machine; the size and roughness are shown in Figure 2b. Tensile tests were performed at room temperature using a wdw-300 micro-controlled electronic universal testing machine. An ICAP7400 plasma spectrometer (Thermo Scientific, Waltham, MA, USA) was used for component detection. In this paper, the chemical composition of the deposit is the average of the measured values at the top, middle and bottom of the deposit. The metallographic specimens were ground and polished to a mirror finish and then etched in mixed acid reagent containing 0.5 vol% HF, 1.5 vol% HCl and 2.5 vol% HNO_3_, with the balance consisting of H_2_O. The etching time is 20 s. A metallographic microscope (OM), scanning electron microscope (SEM) and energy spectrum (EDS) were used to observe the microstructure and analyze its composition. The width of the deposition body was measured by the Vernier caliper. Three parallel samples were taken and the average values of the measurement results were obtained.

## 3. Results and Discussion

### 3.1. Surface Oxidation and Deposition Geometry

Figure 3 shows the surface morphologies of deposits with different Mg contents. It can be seen that, with the increase of Mg content, the surface colour of the deposits gradually deepens, and the lines of deposits on the surface of the deposition body are clearly visible. When the Mg content is less than 7%, the surface texture is smooth, and when the Mg content is 8%, the deposition layer shows a “wave” shape. This indicates that the increase of Mg content will increase the surface oxidation degree. The “wave”-shaped layer is caused by the increasing viscosity of the molten pool, reducing its fluidity when the Mg content is too high [13].

Figure 4 shows the width of deposition bodies with different Mg contents. Due to the increase of Mg content, the viscosity of the molten pool increases, the fluidity and spreading property become worse, and the width of the deposition body slightly decreases; the width of the deposition body decreases by 6.7% when the Mg content is 8% compared with 5%.

### 3.2. Burn Loss Rate of Mg Elements

Figure 5 is the comparison diagram of the Mg burning loss in deposition bodies with different Mg contents. The burn loss rate is calculated by Equation (1):(1)$$A=\frac{X_{1}-X_{2}}{X_{1}}$$
where A is the burn loss rate of the element, $X_{1}$ is the measured element content in the wire, and $X_{2}$ is the measured element content in the deposit.

It can be seen that the deposition bodies all show different degrees of Mg burning loss. When the Mg content increased from 5% to 6%, the burning rate of Mg increased slightly. When the content of Mg was more than 6%, the burning rate of Mg increased significantly. This is because Mg is more active, and its activity is proportional to its concentration.

### 3.3. Microstructure

Figure 6a–d shows the pores in the deposition bodies with different Mg contents. It can be seen that the pores in the deposition body are all round. With the increase of Mg content, the number of pores first decreases and then increases, and the size first decreases and then increases. When the Mg content is 6%, the number and size of pores are the least. There are three stages in the formation of pores: nucleation of bubbles, growth of bubbles and rise of bubbles. The nucleation probability of bubbles depends on Equation (2) [14]:(2)$$j=Ce^{-\frac{4\pi r\sigma }{3KT}}$$
where j is the number of bubble nuclei formed per unit time, r is the critical radius of the bubble core, and σ is the surface tension between the bubbles and the metal liquid. K is the Boltzmann constant (K = 1.38 × 10 − 16 erg/K). T is kelvin (K), and C is constant. When the Mg content increases, the viscosity of the molten pool increases, and the surface tension between the bubbles and metal liquid increases. It can be seen from Equation (1) that the nucleation probability of the bubbles decreases. In addition, the growth of bubbles needs to satisfy the relational Equation (3) [14]:(3)$$P_{h}>P_{o}$$
where $P_{h}$ is the internal pressure of the bubble and $P_{o}$ is the external pressure of the bubble. It can be seen from Equation (2) that, when the Mg content increases, the viscosity of the molten pool increases and the external pressure $P_{o}$ of bubbles increases, which hinders the growth of bubbles.

From the above analysis, it can be seen that increasing the Mg content will reduce the nucleation probability of bubbles and hinder the growth of bubbles, which is why the number and size of bubbles decrease when the Mg content increases from 5% to 6%. However, when the Mg content continues to increase, the number and size of pores increase, which is related to the origin of pores. Hydrogen is the main reason for the existence of pores in the WAAM aluminum alloy. Water in the arc column atmosphere and water absorbed by the wire and substrate are important sources of hydrogen. Mg has high activity and is easy to oxidize. When the content of Mg is increased, the following two sources of hydrogen will be increased: first, the oxide film on the surface of the wire thickens and increases the water absorption; second, during the processing of WAAM, the deposition body surface forms Al_2_O_3_ due to the oxidation and burning loss of Mg, which easily absorbs water in the air. This can be easily seen in Figure 3 and Figure 5. When the Mg content is more than 6%, the surface oxidation is serious, and the burning rate of Mg increases significantly, greatly increasing the water absorption of the body. Therefore, when the Mg content is higher than 6%, the number and size of pores increase.

Cracks are clearly visible in Figure 6c,d, while no obvious cracks are found in Figure 6a,b. Figure 6e is a magnification of cracks in Figure 6d. It can be seen that cracks occur and develop along grain boundaries, and the main extension direction is perpendicular to the deposition direction, which belongs to the crystal thermal crack, which has a serious impact on the vertical mechanical properties. When the Mg content of the alloy is increased, the viscosity of the molten pool increases and the fluidity is poor. During the rapid solidification process of the molten pool, it cannot feed in a timely manner, and the thermal crystallization cracking occurs due to the influence of tensile stress.

Figure 7 shows the microstructure of layers and interlayers (remelting parts) of depositions with different Mg contents. As can be seen, the layer and interlayer tissues can be clearly distinguished. When the Mg content is 5% and 6%, the microstructures of the layers and interlayers are finely equiaxed crystals, and the interlayer grains are smaller. With the increase of Mg content, the grains grew gradually. When the Mg content was 8%, columnar crystals with la arger size appeared in the interlayer. With the increase of Mg content, the number of precipitated phases increased along with the precipitation phase aggregation phenomenon.

Figure 8 shows the two main precipitated phases in the deposition body. According to spectrogram A and spectrogram B, the structure and image of the aluminum alloy [15] and the binary phase diagrams of Al–Mg and Al–Mn show that the two precipitated phases are the (FeMn)Al_6_ phase and β(Mg_2_Al_3_) phase, respectively. The (FeMn)Al_6_ phase is insoluble, hard and brittle, and is a thick sheet segregation polymer. The β(Mg_2_Al_3_) phase is face-centered cubic and brittle at room temperature, meaning that the more of this phase the alloy has, the less plastic it is. Al–Mg alloy is a solid solution strengthening alloy [15]. Parts of the Mg elements are dissolved in the α(Al), and the rest are precipitated in the form of β(Mg_2_Al_3_) phase. Figure 9 shows the contents of Mg and Mn within the grains. As can be seen, with the increase of Mg content, the β(Mg_2_Al_3_) phase is precipitated more. In addition, with the increase of solid solution of Mg element in the α(Al), the Mn solid solution decreases, thus increasing the precipitation of the (FeMn)Al_6_ phase. The precipitation of these two phases results in the coarsening of deposition and the growth of grain.

### 3.4. Mechanical Properties

Figure 10 shows the horizontal and vertical mechanical properties of depositions with different Mg contents. As shown in Figure 10a, when the Mg content in the deposition body is less than 7%, the horizontal tensile strength and yield strength increase with the increase of Mg content. When the Mg content was more than 7%, both the tensile strength and yield strength decreased. The elongation decreases with the increase of Mg content. The strengthening mechanism of the Mg element is solid solution strengthening; therefore, with the increase of Mg content, the content of the solid solution Mg in α(Al) increases, and the tensile strength and yield strength increase. However, when the Mg content was more than 7%, the tensile strength and yield strength decreased due to the coarsening of the tissues, the increase of pores and the severe thermal cracks in the crystals. The decrease of elongation was mainly caused by the increase of the precipitated phase and the thermal cracking of the crystal.

As shown in Figure 10b, the vertical tensile strength, yield strength and elongation all peak when the Mg content is 6%. The mechanical properties of a Mg content greater than 6% were significantly reduced—especially the elongation. The increase of Mg content can improve the vertical mechanical properties, but when the Mg content is more than 6%, the occurrence of thermal cracks causes a sharp decline in the vertical mechanical properties.

Comparing the horizontal and vertical mechanical properties of Figure 10a,b, when the Mg content is 5–6%, the horizontal and vertical properties are more uniform. The difference was smallest when the Mg content was 6%. When the Mg content was more than 6%, the difference in mechanical properties between the horizontal and vertical directions increased significantly. There are two reasons for this difference in mechanical properties: one is the heterogeneity of the layer and the interlayer tissues. The grains of the layer were larger than those in the interlayer, and with the increase of Mg content, β(Mg_2_Al_3_) and (FeMn)Al_6_ phase segregated and aggregated in the layer. The nonuniformity of the layered structure is detrimental to the vertical mechanical properties. The second reason for this difference is the generation of thermal cracking. It can be seen from Figure 6c,d that the extension direction of the crack is perpendicular to the deposition direction—that is, parallel to the layer—which can seriously reduce the vertical mechanical properties.

A comprehensive data analysis shows that when the Mg content is 5–6%, the mechanical properties are better. When the Mg content is 6%, the comprehensive mechanical performance is the best, the horizontal tensile strength, yield strength and elongation are 310 MPa, 225 MPa and 17% respectively, and the vertical tensile strength, yield strength and elongation are 300 MPa, 215 MPa and 15%, respectively. The mechanical property data are consistent with the trend of the microstructure.

### 3.5. Fracture Behaviour

Figure 11 shows the fracture morphology of the tensile specimen. Letters a, b, c and d show the fracture morphology of the horizontal tensile specimen, while e, f, g and h are fracture surfaces of the vertical tensile specimen. It can be seen that all fracture mechanisms are ductile fractures with obvious dimples. When the Mg content was 5% and 6%, the dimples were fine and uniform, and the horizontal and vertical fracture morphology was basically the same. When the Mg content increased to more than 7%, the number of dimples decreased, and more “grape grain”-like loose tissues appeared due to delayed supplementation, especially vertical fractures. This kind of shrunk loose tissue appeared in a large area, corresponding to the crack in Figure 6c,d, which further explained the cause of the crack.

## 4. Conclusions and Future Prospects

(1)The content of Mg affects the surface oxidation degree and geometric size of WAAM Al–Mg alloy deposits. The surface oxidation degree increased with the increase of Mg content. When the Mg content reached 8%, a “wave”-shaped deposition layer appeared.(2)The effect of Mg content on the mechanical properties of the WAAM Al–Mg alloy is significant. The mechanical properties were excellent when the Mg content was controlled at 5–6%. When the Mg content is 6%, the comprehensive mechanical properties were optimized, with the horizontal tensile strength, yield strength and elongation being 310 MPa, 225 MPa and 17% respectively, and the vertical tensile strength, yield strength and elongation being 300 MPa, 215 MPa and 15%, respectively.(3)The effect of Mg contents on the properties of WAAM Al–Mg alloy deposits is mainly attributed to three points: first, with the increase of Mg content, the number of pores first decreases, then increases, and the size first decreases, then increases. When the Mg content is 6%, the number of pores is the least and the size is the smallest. Secondly, when the Mg content reaches 7% or above, a serious shrinkage will appear due to the poor fluidity of the molten pool, which will lead to crystallization heat cracking. Third, with the increase of Mg content, the precipitated phase (FeMn)Al_6_ and β(Mg_2_Al_3_) increased, and the grain size increased, and larger columnar crystals appeared in the layer when the Mg content was 8%.

In this paper, the influence of Mg content on WAAM Al–Mg alloy deposits was systematically described, and the optimal Mg content range was obtained, which is of guiding significance for the development of WAAM Al–Mg alloys. The WAAM technology’s technological characteristics determine that the composition of an alloy fabricated by WAAM is special. In future studies, other alloy elements of Al–Mg alloys will be further investigated to finally obtain the composition range of Al–Mg alloys suitable for the WAAM process, in order to promote the engineering application of WAAM Al–Mg alloys.

## Figures and Tables

**Figure 1 materials-12-04160-f001:**
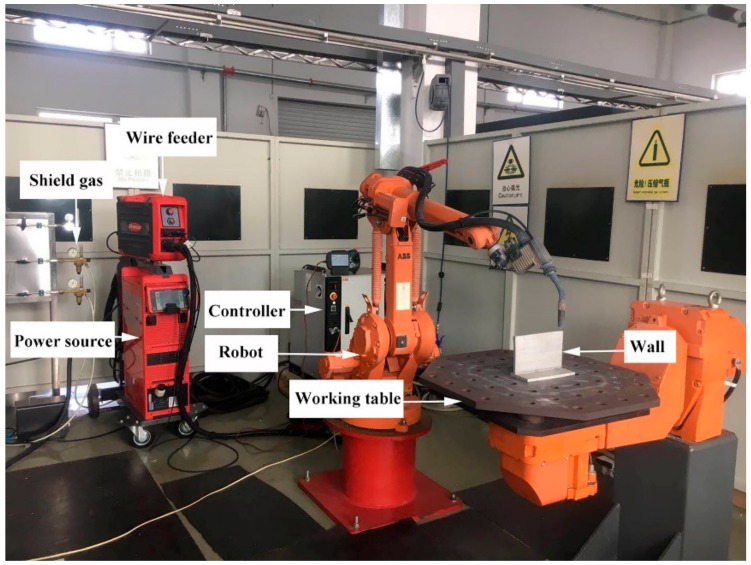
Cold Metal Transfer (CMT)-wire arc additive manufacture (WAAM) system.

**Figure 2 materials-12-04160-f002:**
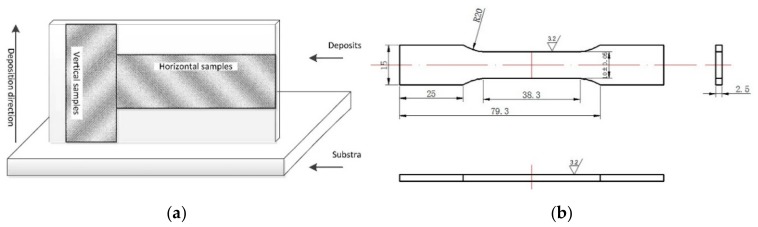
Schematic diagram of extracted tensile sample ((**a**): Sampling position of tensile samples; (**b**): Tensile sample specification (the units for coupon dimension are mm)).

**Figure 3 materials-12-04160-f003:**
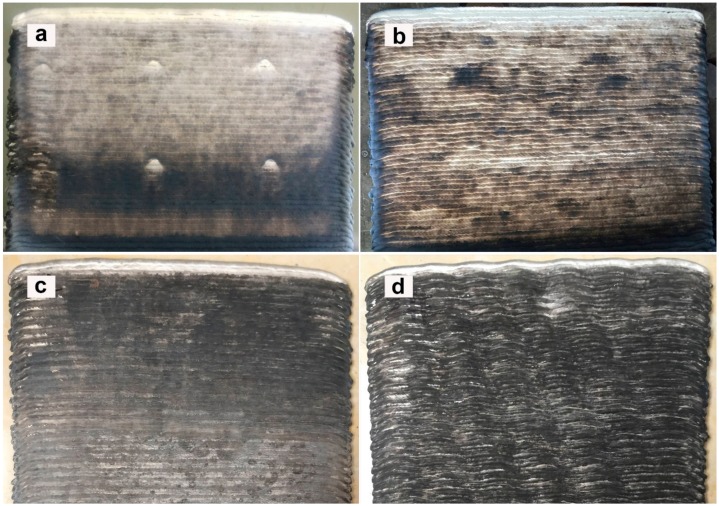
The surface oxidation appearance of WAAM Al–Mg deposits with different Mg contents ((**a**): 5% Mg; (**b**): 6% Mg; (**c**): 7% Mg; (**d**): 8% Mg).

**Figure 4 materials-12-04160-f004:**
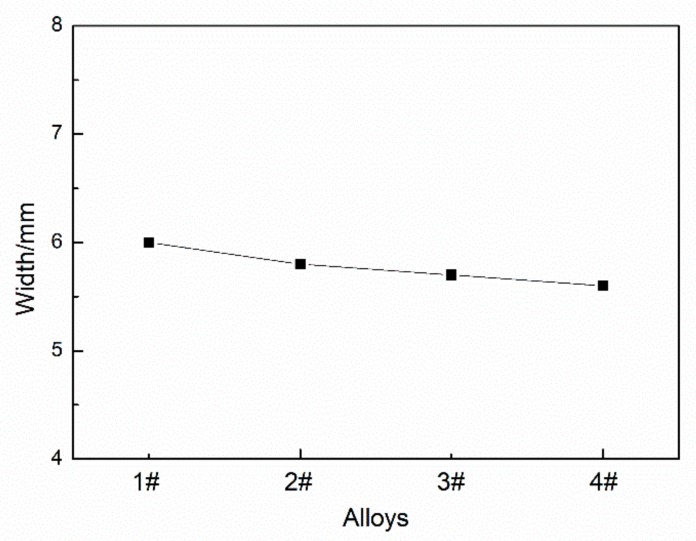
Width of deposits with different Mg contents.

**Figure 5 materials-12-04160-f005:**
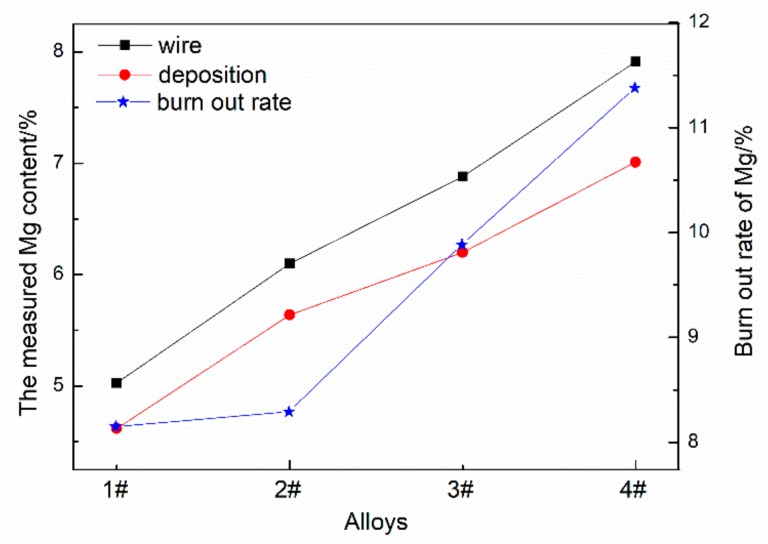
The burning loss rate of Mg in deposits with different Mg contents.

**Figure 6 materials-12-04160-f006:**
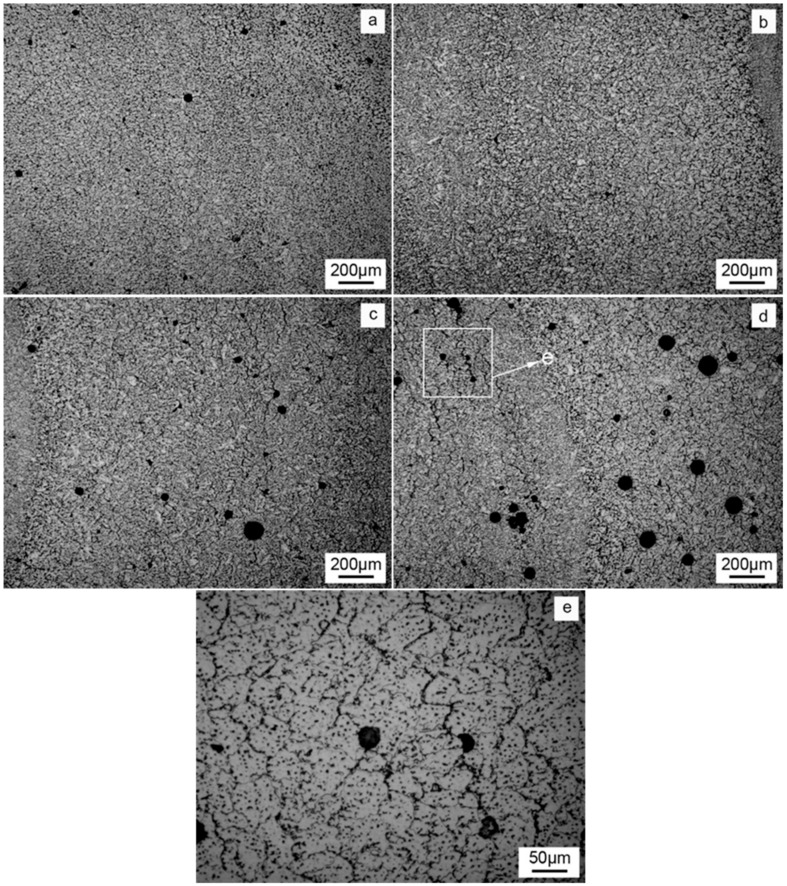
Optically observed porosity and cracking for the deposits with different Mg contents ((**a**): 5% Mg; (**b**): 6% Mg; (**c**): 7% Mg; (**d**): 8% Mg; (**e**): enlarged view of the crack in Figure 6d).

**Figure 7 materials-12-04160-f007:**
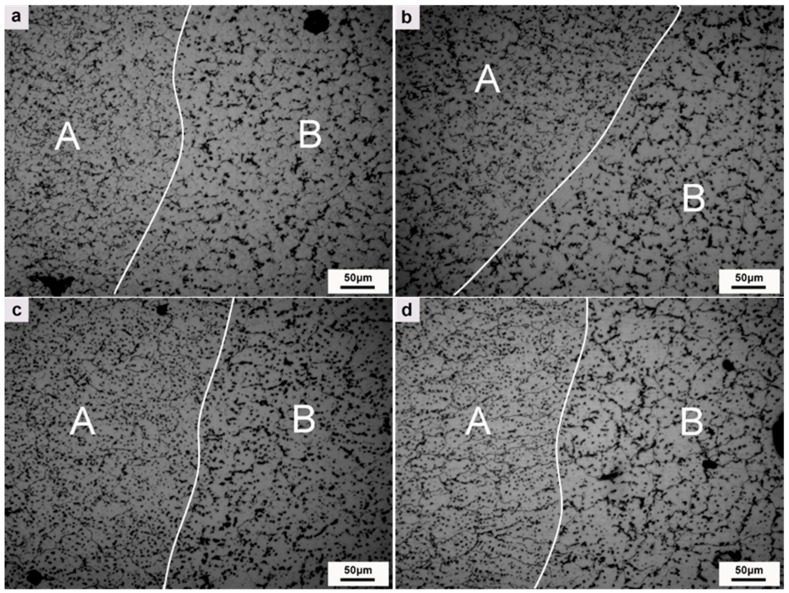
The metallographic structure of deposits with different Mg contents ((**a**): 5% Mg; (**b**): 6% Mg; (**c**): 7% Mg; (**d**): 8% Mg; A: interlayer; B: layer).

**Figure 8 materials-12-04160-f008:**
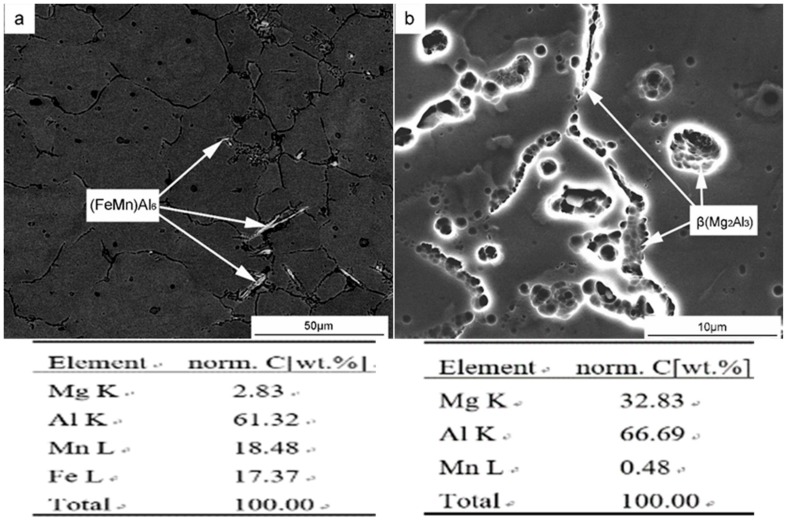
The morphology and composition of the precipitated phase in WAAM Al–Mg alloy deposits ((**a**): (FeMn)Al_6_ phase; (**b**): β(Mg_2_Al_3_) phase).

**Figure 9 materials-12-04160-f009:**
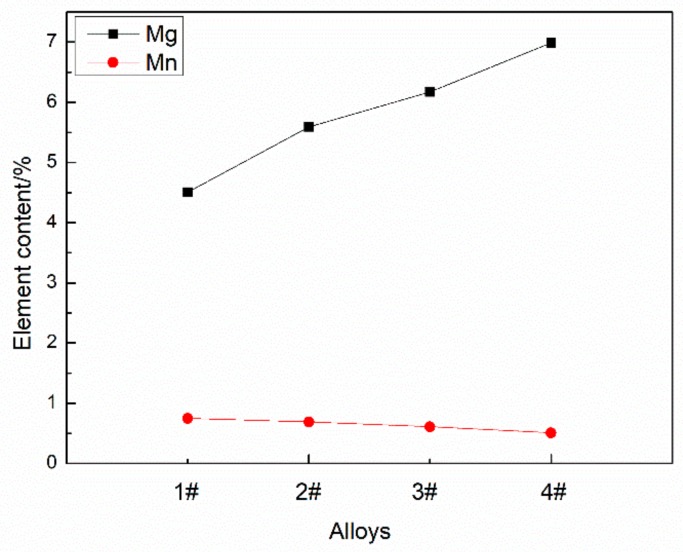
The contents of Mg and Mn within the grains.

**Figure 10 materials-12-04160-f010:**
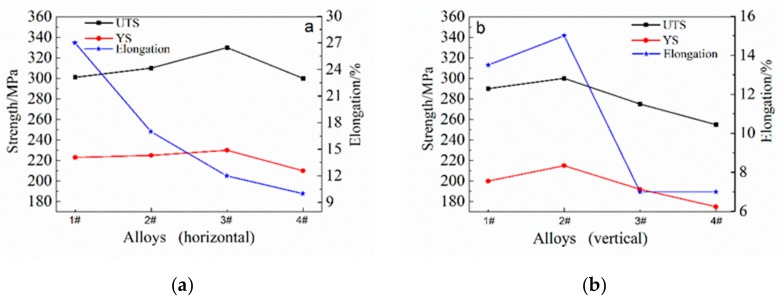
The mechanical properties of deposits with different Mg contents. (**a**): horizontal mechanical properties; (**b**): vertical mechanical properties.

**Figure 11 materials-12-04160-f011:**
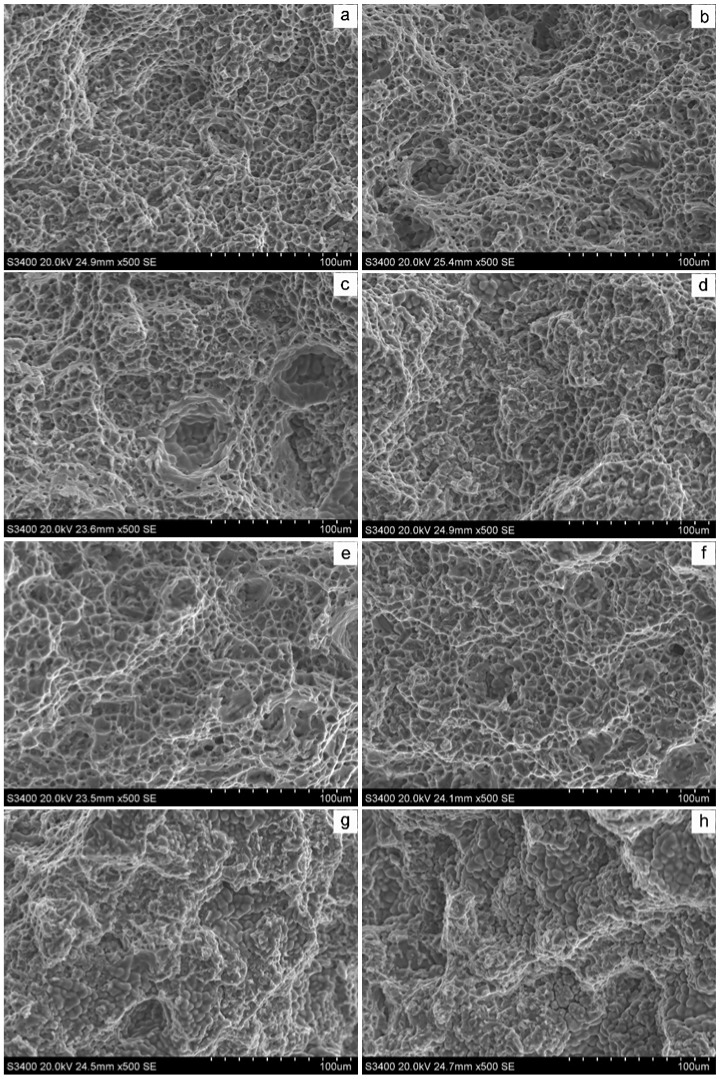
The fracture morphology of tensile simples with different Mg contents ((**a**): 5% Mg; (**b**): 6% Mg; (**c**): 7% Mg; (**d**): 8% Mg; (**e**): 5% Mg; (**f**): 6% Mg; (**g**): 7% Mg; (**h**): 8% Mg. (**a**–**d**): horizontal fracture; (**e**–**h**): vertical fracture).

**Table 1 materials-12-04160-t001:** Chemical composition of welding wire with different Mg contents.

	Si	Fe	Cu	Mn	Mg	Zn	Zr
1#	0.046	0.107	0.0034	0.84	5.03	0.0083	0.098
2#	0.049	0.102	0.002	0.85	6.10	0.0073	0.092
3#	0.041	0.120	0.0019	0.85	6.88	0.0074	0.087
4#	0.042	0.137	0.0063	0.84	7.91	0.0098	0.087

**Table 2 materials-12-04160-t002:** Chemical composition of deposits with different Mg contents.

	Position	Si	Fe	Cu	Mn	Mg	Zn	Zr
1#	Upper	0.047	0.114	0.0040	0.83	4.67	0.0093	0.087
	Middle	0.055	0.113	0.0044	0.79	4.58	0.0096	0.090
	Lower	0.048	0.109	0.0045	0.78	4.61	0.0087	0.087
	**Average**	**0.050**	**0.112**	**0.0043**	**0.80**	**4.62**	**0.0092**	**0.088**
2#	Upper	0.052	0.105	0.0030	0.81	5.68	0.0083	0.086
	Middle	0.057	0.109	0.0029	0.85	5.65	0.0087	0.088
	Lower	0.050	0.113	0.0025	0.80	5.59	0.0082	0.078
	**Average**	**0.053**	**0.109**	**0.0028**	**0.82**	**5.64**	**0.0084**	**0.084**
3#	Upper	0.042	0.131	0.0028	0.74	6.18	0.0074	0.079
	Middle	0.043	0.124	0.0023	0.82	6.23	0.0079	0.080
	Lower	0.047	0.126	0.0027	0.81	6.19	0.0078	0.084
	**Average**	**0.044**	**0.127**	**0.0026**	**0.79**	**6.20**	**0.0077**	**0.081**
4#	Upper	0.050	0.141	0.0062	0.78	7.00	0.0094	0.077
	Middle	0.045	0.138	0.0063	0.85	6.97	0.0099	0.082
	Lower	0.043	0.145	0.0070	0.80	7.06	0.0086	0.078
	**Average**	**0.046**	**0.141**	**0.0065**	**0.81**	**7.01**	**0.0093**	**0.079**

**Table 3 materials-12-04160-t003:** Deposition process parameters.

Process Parameters	
Current	90 A
Arc voltage	10 V
Travel speed	8 mm/s
Wire feed speed	5.5 mm/min
Interlayer wait time	90 s
99.999% argon flow rate	25 L/min

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
