# Peer review of "Effect of Mg Content on Microstructure and Properties of Al–Mg Alloy Produced by the Wire Arc Additive Manufacturing Method"

_materials, 2019, doi:10.3390/ma12244160_

Round 1
Reviewer 1 Report
There is no information in the article on how many samples were produced, are the results obtained repeatable? - it is necessary to include such information in the article.
Fig. 6. Photographs of the microstructure are of rather poor quality, better would be done using SEM in both SE and BSE modes where grain boundaries and phase distribution are clearly visible.
In addition, the caption for Fig. 6 should include the description for Fig. 6e
Fig. 7 – the quality of the microstructures presented in Fig. 7 are unsatisfactory, it would be better to insert better ones, made with the use of the SEM microscope, or using the etching method that allows grains to be visible in the microstructure - then the microstructure of the individual layers would be clearly visible.
In addition, check if the Mg is evenly distributed within the grains - a distribution map of the elements and point analysis within the grain will be helpful.
Fig. 8 - a more detailed analysis should be provided, The microstructure and the point analysis presented in the Fig. 8 is rather poor quality, and didn’t show any diversity for individual phases - it looks like pores rather than precipitates. In addition, it is necessary to present an analysis from the matrix area.
The description of the mechanical properties requires refinement, the explanation of the differentiation of the "horizontal" and "vertical" properties is too laconic.
Fig. 10 - the description should be corrected and taken into account that part of the fractures is "horizontal" (Fig. 10 a, b, c, d) and part "vertical" (Fig. 10 e, f, g, h).
The article would be beneficial if the authors referred the obtained results to those already available in the literature dealing with aluminum alloys.
Reviewer 2 Report
1.The term "wave deformation" for metals is usually used for shock wave propagation and high speed deformation. The same term is used in hydrodynamics. It is not clear for me how the wave form of the layers in the material was obtained. Many reasons should be taken in the account. For example: deformation during crystallization or during cooling, not uniform distribution of Mg concentration during crystallization. Unfortenately this problem was not discussed in a manuscript. I suppose that it will be better to speak not about wave propagation, but about the wave form of the layers obtained.
2. It is necessary to explain terms CMT and GTAW on the page 1.
3. It is necessary to explain how were prepared the surfaces before tensile experiments and before the structure investigation.
4. How was the Mg concentrations measured for Fig.4,5? It is highly recommended to show the errors estimations and the role and the level of Mg concentration inhomogeneity.
5. It will be interesting to estimate the mass losses of Mg for Fig.4 and to compare these with Fig.5 data. Are these data in a good accordance?
6. It will be correct to present Mg, Al and Mn concentrations for Fig.8 in a numeric form as well as to mark the points of analysis.
7. On the page 9 there is a possible typo: an extra number 1 at the beginning of the sixth line.
8. There is a lack of information concerning possible Mg segregation and about the oxides presence in a sample.
Reviewer 3 Report
The authors present a nice study of the effect of Mg additions to Al welding wire for use in wire arc additive manufacturing (WAAM), in an attempt to determine the optimal mass percent Mg for improved mechanical properties of the resultant deposition. Additive manufacturing in general, and of metals in particular, is of great current fundamental and applied interest. With respect to 3D printed metal objects, questions abound regarding their mechanical properties relative to traditional manufactured objects (e.g., cast, etc.), especially with regards to anisotropy in the build direction. While the authors have provided a great deal of data, and determined the optimal Mg content (to within 1%) for their given Al feed wire composition, a few questions remain regarding the experimental methods and the interpretation of the results:
How were the chemical compositions listed in Table 1 determined? What is the effect of varying the process parameters listed in Table 2 (e.g., current, arc voltage, travel and wire feed speeds, wait time, etc.) on the end product (e.g., morphology, mechanical properties, etc.)? How were the parameters decided upon/chosen? Were they optimized in any way? Would different parameters be optimal for different Mg concentrations given the changes in properties (e.g., viscosity of the molten pool, etc.)? How were the test samples (dogbones) "extracted" from the printed deposition body (p. 3)? I'm not familiar with the "burn out" rate and how this is measured (Fig 5 and accompanying text). Please elaborate. Fig 6e (zoom in from Fig 6d) could use a bit of explanation/callout in the Fig 6 caption. It is not immediately obvious from the captions for Fig 6-8 and Fig 10 what the viewing direction is relative to the print/deposition axis; this would be useful to include in the caption and/or label on the figures. It wasn't clear to me exactly where the EDS spectra shown in Fig 8b were obtained from in the SEM micrograph shown in Fig 8a since there are 3 arrows pointing towards each general area. A box indicating the sampling volume would perhaps be more useful/informative. An EDS map of Al, Mg, Mn, and Fe content would also be instructive. More details regarding the SEM instrument and imaging parameters would be helpful (e.g., accelerating voltage, etc.) For Fig 10, highlighting visually somehow the differences between panels a, b, e, and f vs c, d, g, and h (morphology for 5-6% vs 7-8% Mg) would be helpful. With respect to the conclusions (p. 10): The first bullet point re: oxidation would be strengthened with some sort of quantification of the surface oxide content. While the 6% Mg sample exhibits the best mechanical properties (in agreement with the qualitative morphological observations from optical microscopy and SEM), the values obtained for the tensile and yield strengths are seemingly only moderately better than those quoted for Ref 11 (Geng, et al.), which were described as "poor properties", so some context re: desired values/ranges for those properties would be valuable. Finally, is it possible to draw any larger conclusions with respect to how to predict the optimal Mg content for a given Al alloy based on the results of this study? This would greatly increase the impact of the current study, by answering the "So what now?" question.
Upon addressing these questions, I believe this paper is ready for publication.
Round 2
Reviewer 1 Report
The article should indicate whether the results obtained are reproducible or relate only to this set of samples.
Photographs of the microstructure are still of rather poor quality (Fig. 6, 7, 8).
Fig. 8 - despite placing additional photos, the fig. is still of poor quality - the fonts are too small.
In addition, the chemical composition analysis would be better if it was done on the non-etched specimen - Fig. 8b presented in the paper shows "excessively etched” areas, not phases (looks rather like porosity).
It is necessary to present a point analysis of the chemical composition (EDS) from the matrix area, then we will receive information about whether the obtained material is homogeneous. The information provided in the article is insufficient.
Author Response
"Please see the attachment.
